# Increased sugar valuation contributes to the evolutionary shift in egg-laying behavior of the fruit pest *Drosophila suzukii*

**Matthieu Cavey**[1]*, **Bernard Charroux**[1], **Solène Travaillard**[1], **Gérard Manière**[2], **Martine Berthelot-Grosjean**[2], **Sabine Quitard**[1], **Caroline Minervino**[1], **Brice Detailleur**[1], **Yaël Grosjean**[2], **Benjamin Prud'homme**[1]*

1 Aix-Marseille Université, CNRS, IBDM, Institut de Biologie du Développement de Marseille, Campus de Luminy Case 907, Marseille, France, 2 Centre des Sciences du Goût et de l'Alimentation, CNRS, INRAe, Institut Agro, Université de Bourgogne, Dijon, France

* matthieu.cavey@univ-amu.fr (MC); benjamin.prudhomme@univ-amu.fr (BP)

**Data Availability Statement:** All relevant data are within the paper and its Supporting Information files.

## Abstract

Behavior evolution can promote the emergence of agricultural pests by changing their ecological niche. For example, the insect pest *Drosophila suzukii* has shifted its oviposition (egg-laying) niche from fermented fruits to ripe, non-fermented fruits, causing significant damage to a wide range of fruit crops worldwide. We investigate the chemosensory changes underlying this evolutionary shift and ask whether fruit sugars, which are depleted during fermentation, are important gustatory cues that direct *D. suzukii* oviposition to sweet, ripe fruits. We show that *D. suzukii* has expanded its range of oviposition responses to lower sugar concentrations than the model *D. melanogaster*, which prefers to lay eggs on fermented fruit. The increased response of *D. suzukii* to sugar correlates with an increase in the value of sugar relative to a fermented strawberry substrate in oviposition decisions. In addition, we show by genetic manipulation of sugar-gustatory receptor neurons (GRNs) that sugar perception is required for *D. suzukii* to prefer a ripe substrate over a fermented substrate, but not for *D. melanogaster* to prefer the fermented substrate. Thus, sugar is a major determinant of *D. suzukii*'s choice of complex substrates. Calcium imaging experiments in the brain's primary gustatory center (suboesophageal zone) show that *D. suzukii* GRNs are not more sensitive to sugar than their *D. melanogaster* counterparts, suggesting that increased sugar valuation is encoded in downstream circuits of the central nervous system (CNS). Taken together, our data suggest that evolutionary changes in central brain sugar valuation computations are involved in driving *D. suzukii*'s oviposition preference for sweet, ripe fruit.

## Introduction

Evolution of host preferences, for feeding or oviposition, often involves changes in sensory responses that promote behavioral tuning to host-specific chemical cues [1–11]. Changes in the central nervous system (CNS) affecting specific genes or wiring patterns have also been identified, but unraveling how they contribute to behavioral shifts is more challenging [1,12–14].

**Funding:** This work was supported by funds from: -French National Research Agency (EvoSugar ANR-19-CE16-0007, MC, https://anr.fr/) -European Research Council under the European Union's Seventh Framework Programme (FP/2007-2013 / ERC Grant Agreement n° 615789, BP) -A*MIDEX project (n° ANR-11-IDEX-0001-02, BP) funded by the « France 2030» French Government program, managed by the French National Research Agency (ANR, https://anr.fr/) -France-BioImaging infrastructure supported by the French National Research Agency (ANR-10-INSB-04-01, call "France 2030", BP, https://anr.fr/) -Centre National de la Recherche Scientifique (CNRS, BP, YG, https://www.cnrs.fr/en) -Université de Bourgogne (YG, https://www.u-bourgogne.fr/) -Conseil Régional Bourgogne Franche-Comte (PARI grant, YG, https://www.bourgognefranchecomte.fr/) -FEDER (European Funding for Regional Economical Development, YG, https://ec.europa.eu/regional_policy/funding/erdf_en) -European Council (ERC starting grant, GliSFCo-311403, YG) -ANR (PEPNEURON, YG, https://anr.fr/) -SATT-Grand Est/Sayens (DrosoMous, YG, MBG, https://www.sayens.fr/) -Burgundy council (ALIMENN, GM, https://www.bourgognefranchecomte.fr/) The funders had no role in study design, data collection and analysis, decision to publish, or preparation of the manuscript.

**Competing interests:** The authors have declared that no competing interests exist.

**Abbreviations:** CNS, central nervous system; EC50, effective concentration 50; GRN, gustatory receptor neuron; PI, preference index; SEZ, sub-esophageal zone.

Natural breeding sites for fruit flies are not very well documented, but they are known to lay their eggs on decaying organic material, presumably among other substrates. *Drosophila suzukii* has evolved a novel preference for laying its eggs in a wide variety of ripe fruits (for instance, strawberries, raspberries, cherries. . .), in contrast to most other *Drosophila* species, such as *D. melanogaster*, which prefer to lay eggs on fermented and rotten fruits. This novel behavior is causing significant damage to the fruit industry as *D. suzukii* spreads around the world [15,16]. *D. suzukii*'s adaptation has involved both morphological evolution of its ovipositor, which allows it to pierce hard fruit skins [17,18], and changes in behavioral responses to mechanosensory and chemosensory cues [19–22]. Chemosensory cues are likely to involve multiple molecules. For instance, *D. suzukii* shows increased oviposition responses to olfactory cues from ripe fruit [19]. Conversely, reduced gustatory responses to bitter compounds have been proposed to alleviate a hypothetical oviposition inhibition by ripe substrates [20]. In addition, an increased behavioral response to fermentation by-products may contribute to the repulsion of *D. suzukii* by fermented substrates [21].

Fruit sugars—mainly glucose and fructose—are common to all ripe fruits and are gradually degraded during fruit decay, making them potential indicators of fruit maturity for oviposition decisions. Interestingly, *D. suzukii* shows a weaker preference than *D. melanogaster* for sugars (glucose, fructose, and sucrose) over plain agarose in oviposition choice assays. This correlates with a reduced expression of some of the sugar-gustatory receptor genes in taste organs and reduced sensitivity of its gustatory receptor neurons (GRNs) to sucrose and fructose compared to *D. melanogaster*. However, GRN sensitivity to glucose is not reduced in *D. suzukii* compared to *D. melanogaster*, despite the weaker behavioral response to this sugar [22]. Thus, it remains unclear whether physiological changes in sugar-GRNs underlie the divergent behavioral responses to sugar in *D. suzukii*. Furthermore, whether and how these physiological differences in sugar-sensing neurons might contribute to the enhanced preference of *D. suzukii* for ripe fruit substrates remains to be addressed.

Here, we evaluate and compare in detail the role of sugar perception in oviposition decisions in *D. suzukii* and *D. melanogaster*. Using behavioral assays and genetic manipulation of sugar gustatory perception, we show that sugar has, in fact, a higher value as an oviposition cue for *D. suzukii* than for *D. melanogaster* and that sugar perception is required to drive *D. suzukii*'s preference for ripe fruit substrates over fermented fruit substrates. Thus, an increased valuation of sugar has contributed to the evolutionary shift in *D. suzukii*. Calcium imaging experiments suggest that increased sugar valuation is not immediately encoded at the level of the sensory neurons. Our data support the idea that changes in the processing of sugar sensory information have played a central role in the behavioral divergence of *D. suzukii*.

## Results

### Sugar has a higher value for *D. suzukii's* oviposition preference on fruit substrates

The fruit maturity stages preferred by *D. suzukii* and *D. melanogaster* used in previous studies are loosely defined as a fruit matures progressively from unripe, to ripe, to overripe, to fermented and rotten. This limits our ability to identify the relevant cues for interspecies differences in substrate preference. We therefore developed a controlled-fermentation protocol using industrial strawberry purée as a starting ripe substrate to which we added yeast and bacteria to deplete fruit sugars and produce fermentation products. We measured the concentration of fermentation markers (sugar, acetic acid, and ethanol) to assess the outcome of our fermentation reaction and found that sugars were effectively depleted (S1A and S1B Fig). When offered the choice of laying eggs on the ripe or fermented substrates, *D. suzukii* and *D.*

*melanogaster* recapitulated the opposite preferences reported for whole fruit [19] (Fig 1A). Both ripe and fermented substrates were acceptable to both species when presented individually (Fig 1B), so the difference in behavior clearly reflects a divergence in *preference* rather than, for instance, repulsion by a substrate in one species. Strikingly, *D. suzukii* was one of the very few species to prefer the ripe substrate among a range of closely and more distantly related species (Fig 1C). We also verified that the divergent preferences were not due to different adaptations to the sugar-rich diet on which the flies were housed, as this could have biased chemosensory responses [23] (S1C Fig).

We then investigated the contribution of specific chemical cues to this behavioral divergence. We focused on fruit sugars, glucose and fructose, which are abundant in ripe fruit, and we chose acetic acid as a fermentation marker because it is produced from sugar degradation and has previously been shown to act as an oviposition cue for *D. melanogaster* [24–26]. We first opposed sugar-alone (a mixture of glucose + fructose at a 1.6% (w/v) concentration, similar to our ripe strawberry substrate) to acetic acid-alone (at a 1% concentration, similar to our fermented substrate). Interestingly, there was a clear behavioral divergence between species: *D. melanogaster* preferred acetic acid to sugar, whereas *D. suzukii* showed the opposite preference (Fig 1D). Thus, sugar and acetic acid alone, when presented at concentrations corresponding to those of the ripe and fermented substrates, respectively, recapitulate quite well the species divergence observed on complex fruit substrates at different stages of maturity. This suggests that these cues may play a determinant role in species preferences on natural substrates.

To test this further, we assessed the relative importance of acetic acid and sugar in oviposition decisions in the ripe versus fermented substrate choice assay. The addition of acetic acid to the ripe substrate induced a preference shift toward the ripe substrate of similar magnitude in both species (Fig 1E). Thus, under these conditions, acetic acid appears to exert a similar weight on oviposition decisions in the 2 species. In contrast, adding sugar to the fermented substrate was sufficient to abolish *D. suzukii*'s preference for the ripe substrate, whereas it did not increase *D. melanogaster*'s preference for the fermented substrate (Fig 1F). Thus, despite presumably important differences in their chemical composition, the ripe and fermented oviposition substrates are of equal value to *D. suzukii* as long as their sugar concentrations are the same, whereas sugar content appears to be unimportant to *D. melanogaster*. These results suggest that sugar plays a particularly important role in the decision of *D. suzukii* and has a higher value relative to the fermented substrate for this species than for *D. melanogaster*. This led us to focus on sugar and investigate its potential role in evolutionary changes in *D. suzukii*'s oviposition behavior.

## Higher oviposition responses to sugar in *D. suzukii*

Our results contrast with a previous report suggesting that sugar perception may not be critical for *D. suzukii*'s oviposition decisions, based on the observation that *D. suzukii*'s preference for sugar was weaker than that of *D. melanogaster* in two-choice assays against plain agarose [22]. We therefore decided to reexamine thoroughly *D. suzukii*'s oviposition responses to sugar in different types of assays. Since the oviposition response of *D. melanogaster* to sugar has been shown to be context dependent and can range from sugar preference to sugar rejection [27,28], we tested different experimental conditions (chamber size and substrate composition). When sugar was opposed to plain agar, both species preferred to lay on the sugar side, and this was true in all 3 different experimental contexts (Fig 2A, raw egg-laying rate data in S2A Fig). Sugar preference was reversed for both species in the presence of fermentation cues (S2B Fig) as previously reported for *D. melanogaster* [29,30]. Consistent with published data [22], we observed weaker preferences for sugar in *D. suzukii* compared to *D. melanogaster* (Fig 2A),

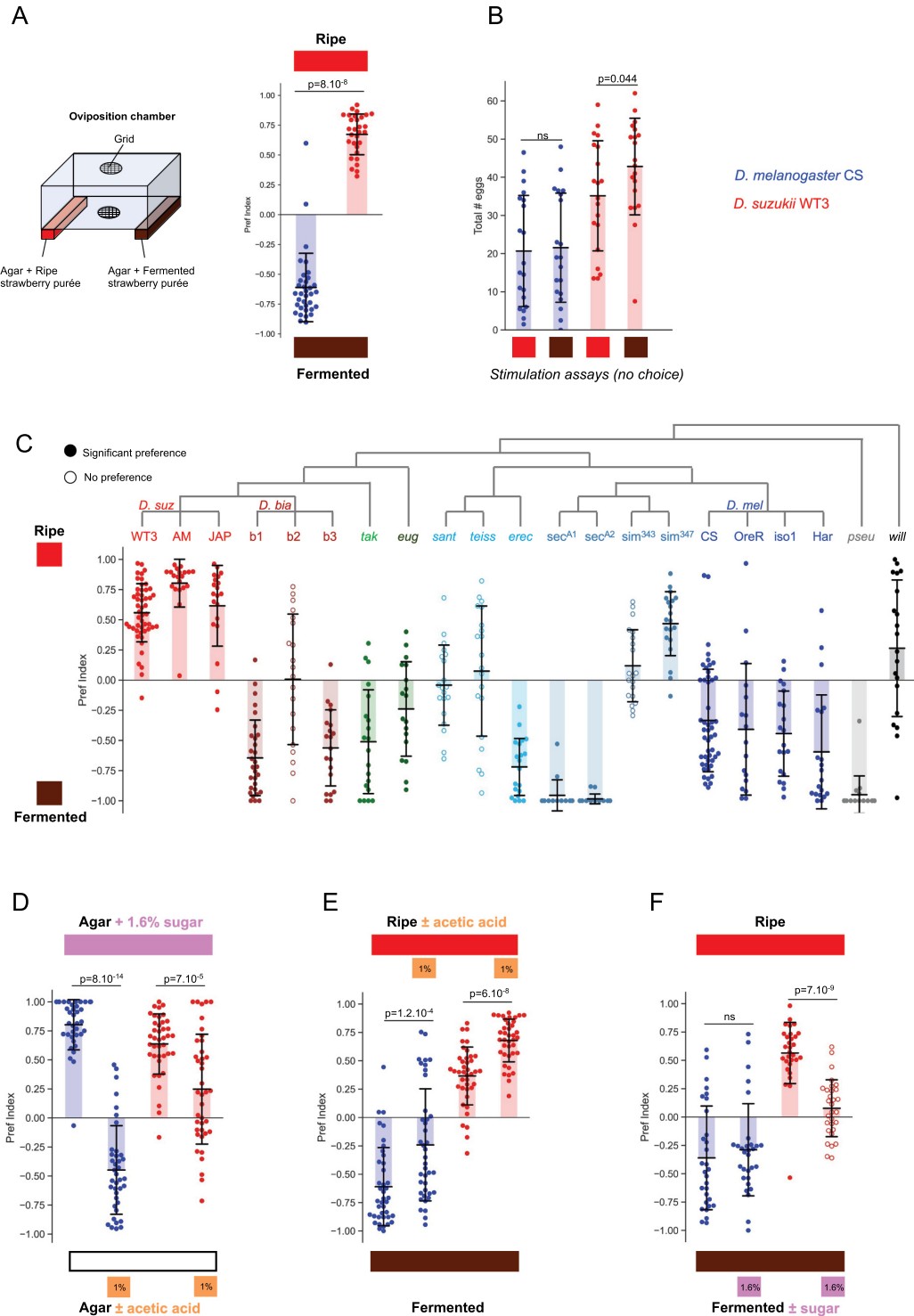

**Fig 1. Sugar is valued more highly by *D. suzukii* in two-choice oviposition assays on natural fruit substrates. (A)** Oviposition substrate preference in a two-choice assay in a large chamber (see Methods for details) opposing a ripe strawberry purée (indicated by the red box above) and the same purée fermented for 3 d under controlled conditions (brown box below; see S1 Fig for additional information and controls). Preference is quantified by a preference index (see Methods). Filled circles in this and the following graphs indicate a significant preference for one of the 2 substrates (Mann–Whitney paired test); open circles indicate no significant preference for either substrate. Shaded bars: mean, error bars:

standard deviation. *D. melanogaster* (blue) and *D. suzukii* (red) show opposite preferences for these substrates. Species preferences are significantly different from each other; Mann–Whitney U test, *p*-values indicated on the graph (*n* = 35, 30 replicates). **(B)** Stimulation (no-choice) assays with these substrates (indicated by red and brown boxes, respectively) show that neither is repulsive to the 2 species, both stimulate oviposition to a similar extent when presented alone (*n* = 20, 20, 20, 20). **(C)** Oviposition assay with ripe vs. fermented substrates for several *Drosophila* species (for some species, multiple wild-type strains were used; see Methods for species names; *n* = 50, 20, 20, 30, 20, 20, 20, 18, 20, 20, 20, 15, 15, 20, 20, 49, 17, 20, 20, 18, 20). **(D)** Two-choice assays opposing sugar alone (glucose + fructose at the concentration found in the ripe substrate, i.e.: 1.6%, indicated by the pink box above the graph) versus agar or agar with acetic acid (1%, similar to fermented substrates, orange boxes below the graph). *D. melanogaster* and *D. suzukii* show opposite preferences for sugar vs. acetic acid (*n* = 36, 39, 40, 40). **(E, F)** Relative value of acetic acid and sugar in the ripe and fermented substrates. **(E)** Adding 1% acetic acid to the ripe substrate shifts oviposition preferences toward acetic acid to a similar extent in both species (*n* = 40, 40, 40, 39). **(F)** Adding 1.6% sugar to the fermented substrate abolishes the preference of *D. suzukii* for the ripe substrate but does not shift *D. melanogaster's* preference for the fermented substrate (*n* = 30, 30, 30, 30). The data underlying this figure can be found in S1 Dataset.

and *D. suzukii* showed poor discrimination ability when faced with 2 different sugar concentrations compared to *D. melanogaster* (Fig 2B). However, we found that *D. suzukii* is naturally stimulated to lay eggs by plain agar, whereas *D. melanogaster* is not (S2A Fig and see below). This effectively reduces *D. suzukii*'s apparent preference for sugar in these assays, complicating the interpretation of the two-choice experiments.

To address this issue, we directly measured the potency of individual sugars in stimulating oviposition for both species using no-choice assays over a wide range of concentrations for the 3 sugars, glucose, fructose, and sucrose. Both species responded positively to sugar in a dose-dependent manner, with *D. suzukii* exhibiting a higher basal egg-laying rate on plain agar, as we have previously noted (Fig 2C and 2D). Strikingly, however, *D. suzukii* began to respond to sugar (i.e., inflection of the curve from basal levels) and reached its maximum oviposition rate at lower sugar concentrations than *D. melanogaster*. Potency can be quantified by the effective concentration 50 (EC50; sugar concentration that induces half-maximal oviposition rate), which corrects for differences in basal oviposition rates. The EC50s were consistently lower for *D. suzukii* compared to *D. melanogaster* for all 3 sugars tested (4- to 10-fold lower). Similar differences were observed with other wild-type strains of these species (Fig 2D). *D. suzukii* is thus more responsive to sugar stimulation than *D. melanogaster*, and these results suggest that its weaker preference for sugar observed in two-choice assays (Fig 2A) may not simply reflect a lower valuation of the stimulus of interest. Furthermore, these results suggest that the lower ability of *D. suzukii* to discriminate between sugar concentrations (Fig 2B) may, in fact, be due to a higher behavioral responsiveness to sugar rather than a lower detection sensitivity.

To further confirm these results, we decided to perform additional two-choice assays using very low concentrations of sugar against plain agar to determine at what concentrations oviposition preference on sugar was first detectable in the 2 species. As there were too few replicates with enough eggs on the lowest sugar concentration substrates, we could not rely on the oviposition preference index. Instead, for each condition, we counted the number of replicates showing a clear positive response to sugar compared to replicates showing either sugar rejection (agar choice) or no choice or no oviposition response (no eggs). Strikingly, *D. suzukii* showed a marked increase in the proportion of positive responses to glucose at lower concentrations than *D. melanogaster* (from 0.05% glucose for *D. suzukii* compared to 0.5% for *D. melanogaster*; Fig 2E, raw data in S2C Fig). These results are consistent with the dose–response experiments and suggest that sugar detection at low concentrations is more likely to trigger oviposition in *D. suzukii* than in *D. melanogaster*.

Taken together, our results suggest that the weaker preference of *D. suzukii* for sugars observed in specific experimental contexts must be interpreted with caution and does not rule

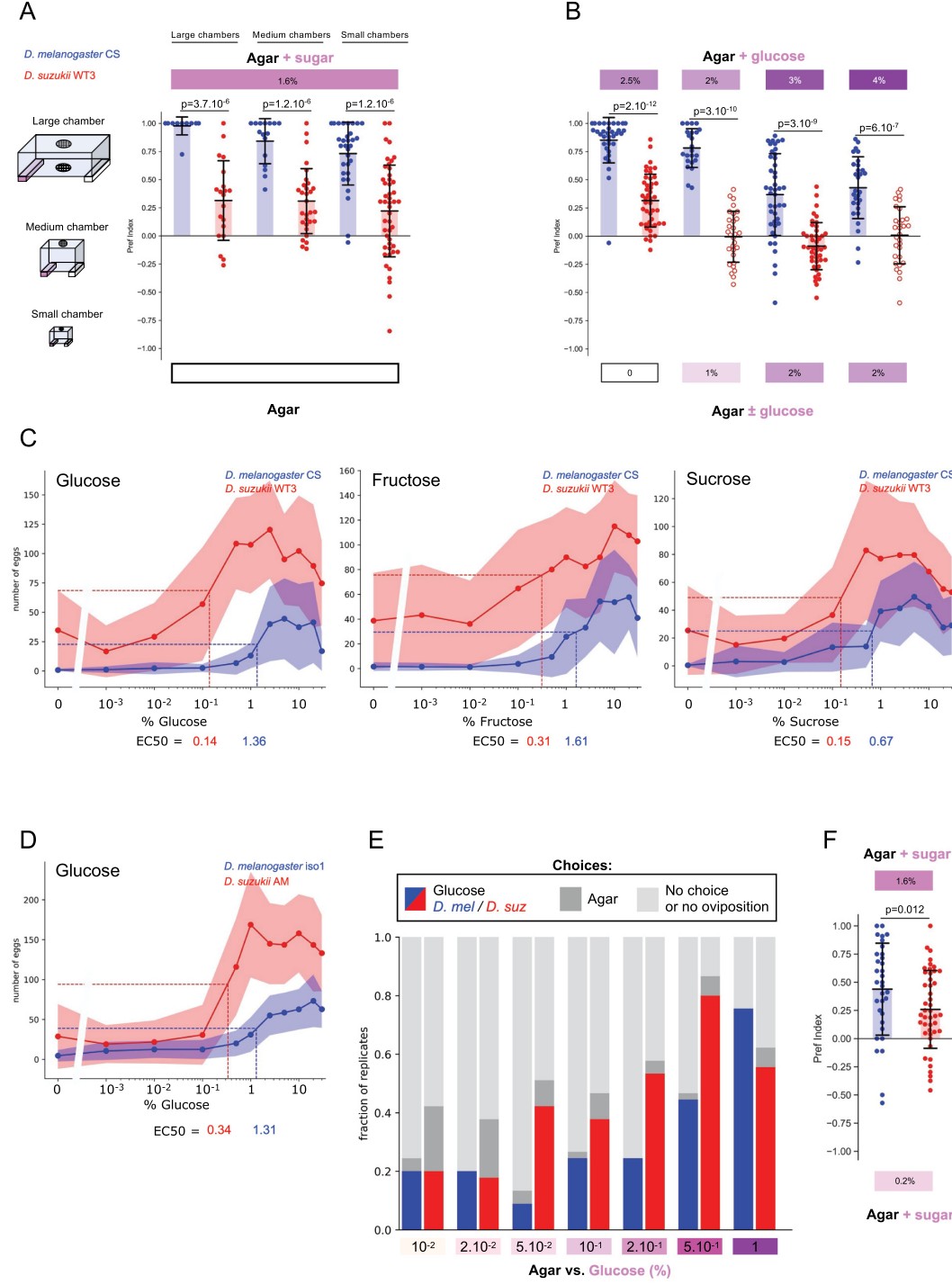

**Fig 2. *D. suzukii* responds to lower concentrations of sugar than *D. melanogaster* in oviposition assays. (A)** Two-choice oviposition assays with sugar at the concentration of the ripe strawberry substrate (1.6% glucose + fructose) versus plain agar (empty box at the bottom) in 3 different experimental setups (see Methods for details). Both wild-type *D. melanogaster* and *D. suzukii* prefer sugar in each experimental setup, but the preference is more pronounced in *D. melanogaster* ($n$ = 12, 20, 16, 30, 31, 48; see raw data in S2A Fig). **(B)** Two-choice assays opposing different concentrations of glucose. *D. suzukii* does not discriminate between concentrations when both are higher than approximately 1% glucose, whereas *D. melanogaster* always shows preference for the higher concentration ($n$ = 35, 45, 23, 30, 43, 44, 29, 30). **(C, D)** Oviposition stimulation assays (no-choice) on increasing concentrations of **(C)** glucose, fructose, and sucrose and **(D)** glucose with 2 other wild-type strains of *D. melanogaster* (iso1) and *D. suzukii* (AM). Data are shown as the mean (dots + lines) +/− standard deviation (shaded areas).

Sugar concentration (x-axis) is on a logarithmic scale. The estimated EC50s are shown at the bottom and with the dashed lines. The EC50 is consistently lower for *D. suzukii* compared to *D. melanogaster* (*n* = 30 for each condition). **(E)** Two-choice assays with low concentrations of glucose versus plain agar. The proportion of replicates choosing glucose (defined as preference index >0.2) is shown by blue/red bars, the proportion of replicates choosing agar (preference index <−0.2) in dark gray, and the proportion with no oviposition response (egg-laying rate <5 eggs) or no choice in light gray. A greater proportion of *D. suzukii* replicates choose sugar at low concentrations (0.05% to 0.5%) compared to *D. melanogaster* (*n* = 45 for each condition; see raw data in S2C Fig). **(F)** Two-choice assay opposing sugar concentrations corresponding to those of the ripe and fermented substrates (1.6% vs 0.2% glucose + fructose, respectively). *D. suzukii* shows a significant preference for the higher concentration substrate (*n* = 33, 45). The data underlying this figure can be found in S2 Dataset.

out a role for this cue in biasing *D. suzukii*'s preference for sugar-rich substrates in natural contexts. In line with this idea, we asked whether *D. suzukii* was able to discriminate between the sugar concentrations present in our ripe (1.6% glucose + fructose) and fermented (approximately 0.2%) strawberry substrates. Indeed, *D. suzukii* chose the higher concentration substrate (Fig 2F) and could therefore, in principle, use this information when choosing between ripe and fermented substrates.

## Sugar perception is required for ripe substrate preference in *D. suzukii*

To formally test the hypothesis that sugar perception is required to guide oviposition choice in *D. suzukii*, we generated genetic tools to manipulate sugar perception in *D. suzukii*. Sugars are sensed in *D. melanogaster* by a family of 9 partially redundant gustatory receptors expressed in approximately 100 GRNs on different body parts and internally (reviewed in [31,32]). We generated a pan-sugar-GRN Gal4 line in *D. suzukii* homologous to the *DmelGr64af-Gal4* line previously shown to drive Gal4 expression in almost all sugar-GRNs [33–35]. Our *DsuzGr64af-Gal4* line labels neurons in the main gustatory organs, the labellum and tarsi, in a highly reproducible pattern reminiscent of *D. melanogaster* (Fig 3A and 3B). Axonal projection patterns in the central brain are also very similar between species (S3A Fig). Neuron counts from our Gal4 line agree very well with those obtained from electrophysiological recordings in *D. suzukii* [22] and collectively show an overall reduction in sugar-GRNs in *D. suzukii* compared to *D. melanogaster* on the proboscis and forelegs (Fig 3C).

Next, we functionally validated our *DsuzGr64af-Gal4* line with a hyperpolarizing *UAS-Kir2.1* transgene [36] we generated in *D. suzukii*. Silencing sugar-GRNs with Kir2.1 significantly reduced sugar preference of both species to a similar extent in two-choice oviposition assays (Fig 3D). However, in both species, sugar preference was not completely abolished, suggesting an incomplete inhibition of sugar perception (which could, for example, be related to insufficient strength of transgene expression). In conclusion, our *DsuzGr64af-Gal4* line appears to target the full complement of sugar-GRNs in *D. suzukii* and produces similar effects in behavioral assays to existing tools in *D. melanogaster*.

We then functionally tested the contribution of sugar sensing to oviposition decisions on complex fruit substrates in *D. melanogaster* and *D. suzukii*. Remarkably, while silencing sugar-GRNs had no effect on *D. melanogaster*'s preference for the fermented substrate over the ripe one, it significantly reduced *D. suzukii*'s preference for the ripe substrate over the fermented substrate (Fig 3E). This manipulation pushed *D. suzukii* close to indifference between the 2 substrates and closer to the *D. melanogaster* state. Since a subset of sugar-GRNs in *D. melanogaster* have been shown to respond to the fermentation product acetic acid, which can act as an oviposition cue [24–26], we asked whether defects in acetic acid perception could contribute to this phenotype. However, silencing of sugar-GRNs did not affect the oviposition responses of *D. suzukii* (or *D. melanogaster*) to acetic acid at concentrations similar to those of our fermented substrate (0.5% to 1%; S3B Fig). In conclusion, our results show that sugar

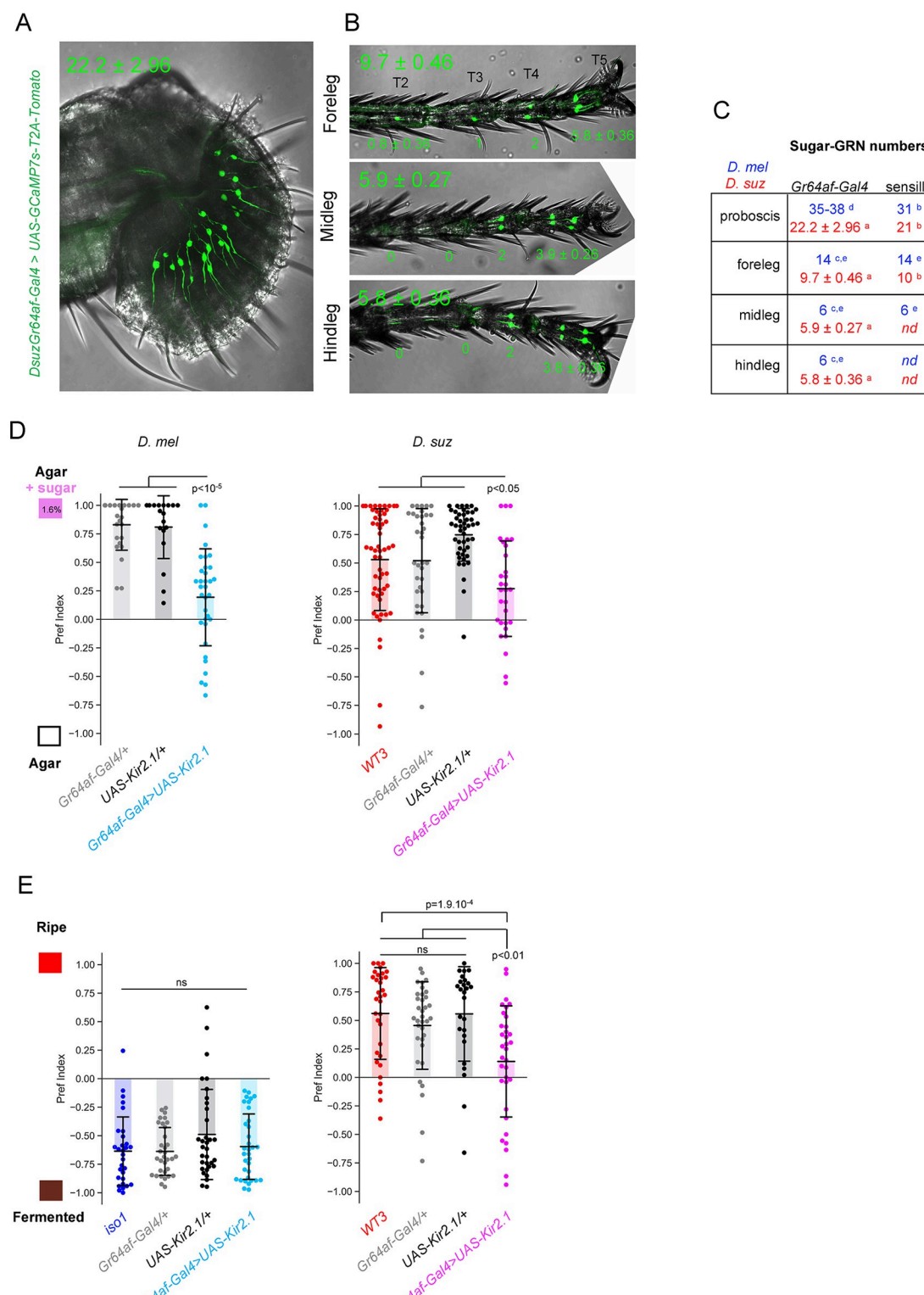

**Fig 3. Sugar sensing is required for ripe substrate preference in *D. suzukii*.** *DsuzGr64af-Gal4* expression in female proboscis (**A**) and legs (**B**) observed with a *UAS-GCaMP7s-T2A-Tomato* reporter. The mean number of positive neurons ± standard deviation is indicated for each organ on the upper side and each tarsus (*n* = 14 proboscises, forelegs, midlegs, and hindlegs imaged; see also S3A Fig for additional characterization of the *DsuzGr64af-Gal4* line). (**C**) Comparative sugar GRN counts in *D. melanogaster* (blue) and *D. suzukii* (red) females determined from *Gr64af-Gal4* reporter expression and electrophysiogical

recordings of sensilla showing a consistent reduction in *D. suzukii* proboscis and forelegs compared to *D. melanogaster* ([a]this study; [b][22]; [c][33]; [d][34]; [e][35]; nd: not determined). **(D)** Two-choice oviposition assay with 1.6% glucose + fructose vs. plain agar. Sugar-GRN inhibition via *UAS-Kir2.1* expression reduces—but does not completely abolish—the response to sugar to a similar extent in both species (*n* = 25, 18, 34, 56, 35, 46, 30). **(E)** Oviposition choice assay on ripe vs. fermented substrate. Sugar-GRN inhibition does not alter the preference of *D. melanogaster* but significantly reduces the preference of *D. suzukii* for the ripe substrate (*n* = 30, 33, 33, 34, 33, 35, 26, 34). The data underlying this figure can be found in S3 Dataset.

perception in *D. suzukii* significantly contributes to oviposition preference toward a ripe substrate, confirming that sugar is one of the major determinants of preference on natural substrates in *D. suzukii*.

## Increased sugar valuation in *D. suzukii* is not encoded in the peripheral nervous system

The increased responsiveness of *D. suzukii* to sugar in behavioral assays suggests evolutionary changes in the processing of sugar information in neural circuits controlling oviposition. Since evolutionary changes have often been observed in sensory neurons [1–11], the divergent behavior of *D. suzukii* could be encoded directly at the level of sugar detection, for instance, via increased GRN responses to low sugar concentrations. However, previous evidence from electrophysiological recordings is rather ambiguous. Specifically, fewer sensilla were found to respond to the 2 sugars sucrose and fructose on the *D. suzukii* labellum than in *D. melanogaster*. However, no differences in sensilla number were observed for responses to glucose on the labellum, and no differences were observed for responses to all 3 sugars on the foreleg [22]. Whether and how a reduction in sensilla responding to 2 of 3 sugars on a sensory organ contributes to behavioral divergence between species remains unknown.

As an independent approach to this study, we therefore decided to use calcium imaging to compare sugar sensory responses across species and generated a *UAS-GCaMP7s* transgenic line in *D. suzukii* for this purpose. We focused on labellar GRNs, for which interspecies differences have been reported, and measured calcium responses in their synaptic terminals in the brain sub-eosophageal zone (SEZ) upon stimulation of the labellum with a range of sugar concentrations (Fig 4A). Sugar stimulation of the labellum has been shown to be sufficient to elicit oviposition in *D. melanogaster* [37].

Both species showed dose-dependent GCaMP responses to glucose, fructose, and sucrose, and interestingly, while the magnitude of the response was generally similar at low sugar concentrations, it was significantly larger in *D. melanogaster* than in *D. suzukii* at higher concentrations (Fig 4B and 4C, raw data in S4 Fig). However, we observed a similar difference upon neuronal depolarization with 1 M KCl (Fig 4C), suggesting that either (i) for technical reasons our ability to detect calcium transients is lower in *D. suzukii*, or (ii) for physiological reasons, calcium influx is always lower in *D. suzukii* GRNs. In favor of the first hypothesis, we noticed that the GCaMP signal-to-noise ratio differed between species, with prestimulation GCaMP fluorescence intensities being lower in *D. suzukii* than in *D. melanogaster* and background fluorescence being higher in the brain. In an attempt to eliminate this potential experimental bias, we normalized the sugar responses to the average KCl responses, so as to express the sugar responses as a function of the maximum detectable responses. This normalization removed interspecies differences in glucose and sucrose responses, but responses to fructose (2.5%) still appeared clearly lower in *D. suzukii* than in *D. melanogaster* (S4B Fig), consistent with electrophysiological data [22].

In conclusion, our data do not support the idea that sugar detection in the Peripheral Nervous System (PNS) is enhanced in *D. suzukii*. If anything, GRN responses to fructose are actually weaker over certain concentration ranges, as previously reported [22]. This suggests

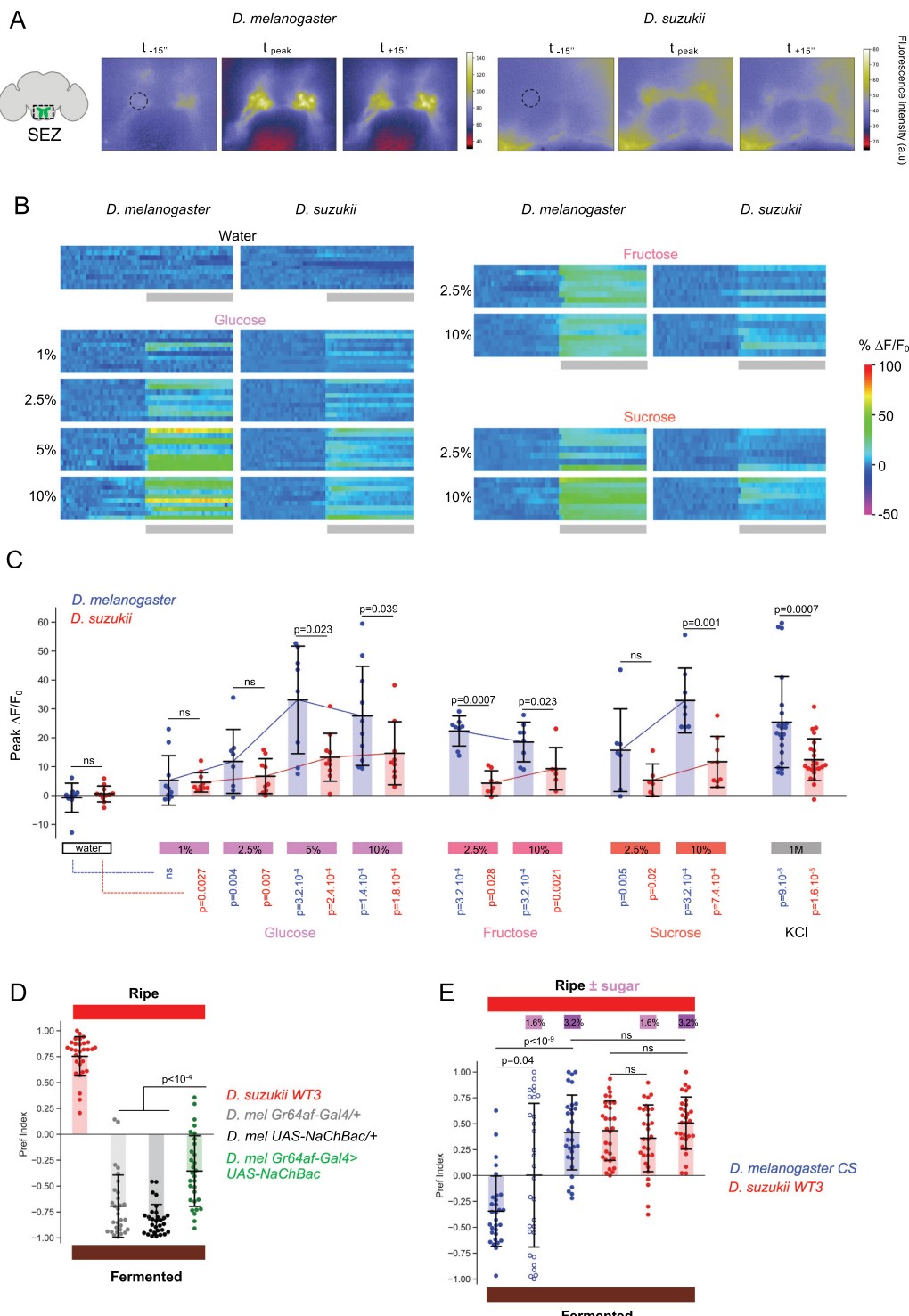

**Fig 4. Comparative calcium imaging of sugar responses in the PNS. (A)** GCaMP7s imaging in the synaptic terminals of sugar-GRNs in the SEZ upon stimulation of the proboscis with glucose. Example images are shown for the 2 species 15 s before stimulation, at the peak response, and 15 s after stimulation. Fluorescence intensity is color coded (scales on the side). ROIs used for quantification are indicated (dashed circles). **(B, C)** Stimulation with water, glucose, fructose, sucrose, and KCl. **(B)** $\Delta F/F_0$ traces of individual ROIs plotted as heat maps (color scale at bottom right) for the different conditions. The period of water or sugar stimulation is indicated by gray rectangles at the bottom. **(C)** Distribution of peak $\Delta F/F_0$ for each condition (shaded area: mean, error bars: standard deviation). The magnitude of the calcium response is generally

higher for *D. melanogaster* at high sugar concentrations (Mann–Whitney U test, *p*-values indicated on the graph). *p*-Values in color below the graph indicate statistical comparisons with water controls (*n* = 9, 11, 10, 10, 8, 9, 8, 10, 10, 9, 8, 7, 8, 6, 7, 6, 8, 8, 22, 23 brains). See **S4 Fig** for additional information. **(D, E)** Manipulation of sugar perception in *D. melanogaster* shifts its oviposition preference in ripe versus fermented substrate assays. **(D)** Increasing sugar input to the CNS via *UAS-NaChBac* expression in sugar GRNs significantly increases the value of the ripe substrate. Compared to parental controls (gray and black), *Gr64af>NaChBac* females (green) show a shift in preference toward the ripe substrate (*n* = 30, 30, 30, 30). **(E)** Increasing the sugar concentration of the ripe substrate increases its value relative to the fermented substrate for *D. melanogaster*. One dose of glucose + fructose (1.6%) added to the ripe substrate (which already contains 1.6% sugar) eliminates the preference for the fermented substrate, whereas the addition of 2 doses (3.2%) reverses the preference in favor of the sweeter ripe substrate (*n* = 30 for each condition). The data underlying this figure can be found in S4 Dataset.

that the increased valuation of sugar for oviposition decisions in *D. suzukii* is likely related to changes in signal processing in the CNS. This would imply that low sugar inputs from the PNS are more potent in inducing oviposition in *D. suzukii*, whereas higher inputs would be required in *D. melanogaster*. One prediction from this model is that *D. melanogaster* should prefer sweet oviposition substrates only when sugar is present at high concentrations. Therefore, it should be possible to bias *D. melanogaster*'s preference for a ripe substrate by artificially increasing the electrical output of sugar-GRNs or by directly increasing the sugar concentration of the substrate. We tested this idea by first expressing the voltage-gated bacterial sodium channel NaChBac [38] in *D. melanogaster*'s sugar-GRNs to hypersensitize them. We confirmed that this manipulation significantly enhanced the oviposition response to sugar in stimulation assays and the calcium responses in sugar-GRNs (S4C and S4D Fig). Consistent with our prediction, this manipulation resulted in a significant reduction in the oviposition preference for the fermented substrate over the ripe substrate (Fig 4D). Second, directly increasing the sugar concentration of the ripe substrate opposed to a fermented substrate also shifted *D. melanogaster*'s preference toward the ripe substrate and even reversed its preference when a high sugar concentration was used (Fig 4E). Sugar is therefore a weak positive oviposition cue for *D. melanogaster*, capable of overcoming the attraction of fermentation cues only when present at very high concentrations. In conclusion, high signaling of sugar-GRNs in *D. melanogaster* leads to oviposition responses and substrate preferences similar to low signaling in *D. suzukii*. These data support the notion that changes in the processing of sugar sensory information in the CNS have contributed to the evolutionary shift of the oviposition host in *D. suzukii*.

## Discussion

Here, we show that sugar has become a key chemosensory cue that guides oviposition decisions in *D. suzukii*. Glucose and fructose, which are abundant in ripe fruits, are a major source of nutrition for insects and can act as attractants for *Drosophila* oviposition in certain contexts [27,28]. We found that *D. suzukii* favors sugar over the fermentation product acetic acid, is more responsive to sugar in several oviposition contexts, and that sugar perception is required for *D. suzukii* to prefer a ripe fruit substrate over a fermented one. In contrast, *D. melanogaster* gives greater weight to fermentation cues such as acetic acid than to sugar, is less responsive to sugar in oviposition stimulation assays, and does not require sugar perception to choose between ripe and fermented substrates in two-choice assays. Collectively, our data suggest that the relative value of sugar as an egg-laying substrate—relative to other compounds, including fermentation products such as acetic acid—has increased in *D. suzukii* compared to other species and that this has, in part, driven the evolutionary shift in substrate preference.

Strikingly, we found that *D. suzukii* can initiate oviposition responses at extremely low sugar concentrations, well below those found in the flesh of ripe fruit, and thus presumably well below what would be required to promote ripe fruit preference. However, the sugar concentration accessible to GRNs on the fruit surface is likely to be lower than in the flesh. Thus,

the extension of *D. suzukii*'s behavioral responses to low sugar concentrations may have played a critical role in promoting ripe fruit preference. In contrast, *D. melanogaster* targets rotten but also occasionally overripe fruits with exposed flesh—and thus high sugar content. These observations are consistent with our results showing that only high sugar concentrations can promote sweet substrate preference in *D. melanogaster*.

Surprisingly, *D. suzukii* showed a weaker strength of preference for sugar compared to *D. melanogaster* in two-choice assays against plain agarose [22] or agar (this study). This observation suggests that sugar perception may not be critical for *D. suzukii*'s oviposition decisions [22], so it is unclear how changes in sugar perception could explain its evolutionary shift toward sweet oviposition hosts. However, our additional experiments—no-choice assays, experiments with natural fruit substrates, and genetic manipulations of sugar perception—show that sugar does indeed play a significant role in *D. suzukii*'s oviposition decisions. In particular, we show that *D. suzukii*'s overall weaker preference for sugar in two-choice assays is most likely due to a high basal egg-laying rate on plain agar. However, when opposing low concentrations of sugar to plain agar, a greater proportion of *D. suzukii* replicates chose the sugar option compared to *D. melanogaster*. This suggests that the likelihood of favoring a given option and the strength of the choice are 2 dissociable parameters that are confounded in the preference index metric.

At the neuronal level, a simple scenario for the increased valuation of sugar in *D. suzukii* would be an increased response directly in the PNS. This could be supported, for example, by an increased number of sweet-sensing GRNs and/or increased sensitivity or firing activity of these neurons. However, our data suggest the opposite:

First, we observed a reduction in the number of sugar-GRNs in *D. suzukii* using our *Gr64af-Gal4* reporter analysis. Our GRN counts agree very well with electrophysiological data [22], and our functional analyses suggest that our reporter does indeed target the full complement of sugar-GRNs. However, we cannot exclude the possibility that there are more subtle differences in the number of functionally distinct sugar-GRN subtypes between species. Indeed, different subpopulations of sugar-GRNs in *D. melanogaster* exert opposite effects on oviposition behavior, with some of the GRNs on the legs inhibiting oviposition, whereas GRNs on the proboscis promote oviposition in a specific context [37]. Interestingly, we observed a reduction in the number of sugar-GRNs in tarsal segments 2, 3, and 4 of the *D. suzukii* foreleg compared to *D. melanogaster*, as well as a reduction on the labellum. This raises the possibility that the disappearance of specific neuronal subtypes in *D. suzukii* may have relieved inhibitory sugar inputs to CNS oviposition circuits.

Second, our calcium imaging analysis did not reveal a higher sensitivity of *D. suzukii*'s sugar GRNs compared to *D. melanogaster*'s. Consistent with electrophysiological data [22], GRN responses to glucose were not higher in *D. suzukii*, and, in fact, responses to fructose were lower than in *D. melanogaster* under some conditions. Taken together, these observations suggest that the behavioral tuning of *D. suzukii* to lower sugar concentrations is due to changes in the processing of sugar sensory information in downstream CNS circuits. We hypothesize that oviposition-triggering circuits in *D. suzukii* can be activated by lower synaptic activity from sugar GRNs than in *D. melanogaster*. This could explain why the maximum oviposition rate in *D. suzukii* is reached at lower sugar concentrations than in *D. melanogaster* and why *D. suzukii* does not discriminate sugar concentrations well in choice assays. Several candidate CNS neuronal groups that receive inputs (directly or indirectly) from sugar gustatory centers and regulate oviposition decisions have been identified in *D. melanogaster*: the TPN2 neurons [37], dopaminergic neurons [29,39], and the oviIN-oviEN-oviDN circuit [40]. Future efforts should aim to identify the functional homologs of these neurons in the *D. suzukii* brain and compare their physiological properties and functional roles across species.

## Materials and methods

### Experimental model and subject details

***Drosophila* stocks.** Flies were reared on homemade Nutrifly medium or standard corn-meal medium where indicated, at 23˚C, 60% relative humidity in a 12:12-h light cycle. Wild-type *D. melanogaster* were Canton S, Canton S SNP iso2 (BL6365), OreR, Harwich (BL4264), and iso1 (isofemale line from India, B. Prud'homme). Wild-type *D. suzukii* were WT3 [41], an isofemale line from Japan (JAP), and an isofemale line from France (AM). Other wild-type species in Fig 1C were (*bia*) 3 isofemale *D. biarmipes* lines from Bangalore, India (b1, b2, b3); (*erec*) *D. erecta* #223; (*sec*) *D. sechellia* A1 (14021–0248.28) and A2 (14021–0248.07); (*sim*) *D. simulans md221#*343 (P. Andolfatto) and *D. simulans Vincennes#347*, obtained from V. Cour-tier-Orgogozo; (*tak*) *D. takahashii*; (*eug*) *D. eugracilis*; (*sant*) *D. santomea*; (*teiss*) *D. teissieri*; (*pseu*) *D. pseudoobscura*; and (*will*) *D. willistoni*.

*D. melanogaster* transgenic lines: *Gr64af-Gal4* (BL57668), *UAS-Cd4tdTomato* (BL35841), *UAS-Kir2.1* [36], *UAS-NaChBac* [38], *UAS-GCaMP7s* (BL80905).

**Transgenic *D. suzukii* lines.** -*DsuzGr64af-Gal4*$^{3xp3GFP}$: A 10,483-bp fragment spanning the *Gr64a* 5′ to *Gr64f* 5′ region from the *D. suzukii* genome was synthesized by Genewiz and cloned into our PiggyBac transformation vector [19]. This fragment is orthologous to the 10-kb fragment used in the *DmelGr64af-Gal4* line [33].

-The *UAS-Cd4tdTomato*$^{3xp3RFP}$ line was described in [19].

-The *UAS-GCaMP7s-T2A-Tomato*$^{3xp3RFP}$ line was generated via PiggyBac transformation of a vector obtained from D. Stern.

-*UAS-Kir2.1*$^{attP2-3xp3RFP}$ line: To facilitate future transgenic approaches, we first generated an attP landing site in *D. suzukii* via CRISPR-mediated homologous recombination, following our CRISPR protocol described in [19]. To design sgRNA oligos, we amplified and sequenced the *D. suzukii* genomic region orthologous to the *D. melanogaster* attP2 locus on chromosome 3. We targeted the following N18GG sequence: AGTTGTTTATAAACTAGGTGG with the sgRNA-F oligo (all primer sequences given below), coupled with the generic oligo sgRNA-R and coinjected with the sgRNA-attP2 oligo to insert the attP site. We screened for transfor-mants by PCR and sequencing, using suzssODNattP2-F1 and suzssODNattP2-R primers for PCR and suzssODNattP2-F2 and suzssODNattP2-R primers for sequencing.

We PCR-amplified the Kir2.1 sequence from the *D. melanogaster* transgenic line using Kir2.1_Fw1 and Kir2.1_Rv1 primers and cloned it by in-fusion into a pBac-attB-UAS vector generated from the PiggyBac vector used in [19]. We then inserted the *UAS-Kir2.1* transgene at the *D. suzukii* attP2 site via the PhiC31 integrase system according to the protocol described in [42]: phiC31 mRNA was produced using the mMessage mMachine kit (Ambion, Austin, Texas) with 1 μg of *Bam*HI-linearized pET11phiC31poly(A) plasmid template. phiC31 inte-grase-capped mRNA was injected into embryos at 1,000 ng/μl, along with 200 ng/μl of UAS--Kir2.1 plasmid.

### Primers

sgRNA-F:

```
GAAATTAATACGACTCACTATAGGAGTTGTTTATAAACTAGGGTTTTAGAGCTA
GAAATAGC
```

sgRNA-R:

```
AAAAGCACCGACTCGGTGCCACTTTTTCAAGTTGATAACGGACTAGCCTTATTT
TAACTTGCTATTTCTAGCTCTAAAAC.
```

sgRNA-attP2:

```
CGGTGGTACTTGTGGGTATATGAAGCCGATTGGAGGGTTCCGACTGCACCCG
GCGAGAGTTGTTTATAAACTAGTAGTGCCCCAACTGGGGTAACCTTTGAGTTCTC
TCAGTTGGGGGCGTAGAATTCAGGTGGAGAGTGGCGATCATATGGACAAGCTTG
CCATTCGCCAAAGCTTTCTAATACCAGTGTTAGTTAGCT
```

    suzssODNattP2-F1: CTTCAGGTAACCGGTTGTGG
    suzssODNattP2-R: ATGCTGCCAAAGCAGTGTCT
    suzssODNattP2-F2: GGGGTACGAGGTGTTTTTAAG
    Kir2.1_Fw1: ATGGGCAGTGTGCGAACCAACCGC
    Kir2.1_Rv1: TCATATCTCCGACTCTCGCCGTAAGG

## Strawberry fermentation

A model rotten/fermented fruit substrate was prepared using frozen organic unsweetened strawberry purée purchased from Sicoly (https://www.sicoly.fr/) as the starting source of ripe fruit. The purée was diluted 1:2 in milliQ water, and 2 g glucose/100 ml was added at the start of fermentation. Fermentation was carried out with 2 microorganisms, the yeast *Saccharomyces cerevisiae* and the acetic acid bacterium *Acetobacter pomorum*. In this system, the 2 microorganisms break down sugar to produce fermentation products including alcohol (yeast) and acetic acid (bacteria). For each fermentation, fresh overnight liquid cultures of *S. cerevisiae* (gift from A. Michelot) and *A. pomorum* (gift from J. Royet) were grown at 30˚C in standard media (YPD and MRS, respectively) and inoculated at a final concentration of $10^6$ cells/ml of diluted strawberry purée. Fermentation was performed in large Erlenmeyer flasks in a shaking incubator for 72 to 96 h at 30˚C. The fermented purée was stored at 4˚C and used over several days/weeks. We optimized inoculation conditions and fermentation duration empirically, using the oviposition choice behavior of wild-type *D. melanogaster* and *D. suzukii* as a guide. This fermentation protocol yields substrates of variable quality, presumably due to differences in the composition of the starting ripe strawberry purée from batch to batch.

Glucose, fructose, acetic acid, and ethanol dosages of fermented and unfermented purées were performed using kits from Megazyme (K-FRUGL, K-ACET, K-ETOH).

## Chemicals

Glucose (sigma G7021), fructose (sigma F0127), sucrose (sigma S1888), acetic acid (Carlo Erba 524520), casein hydrolysate (sigma 22090), agar (vwr 20768.361).

## Behavioral experiments

**Experimental setups.** The experiments were performed in custom-built setups—"large," "medium," and "small" chambers—adapted from [43], except the sugar dose–response experiments of Fig 2C and 2D, which were performed in the experimental setup described in [19]; see below. The large, medium, and small setups consisted of sheets of drilled Plexiglas assembled to form large (6 × 4 × 1.8 cm), medium (3 × 2 × 1.8 cm), or small (1 × 1.2 × 1.8 cm) chambers each containing 2 strips of oviposition substrate at the bottom at opposite ends of the chamber (0.8 cm wide for large and medium chambers, 0.3 cm wide for small chambers; see schematics in Fig 2A). Three females were introduced in large and medium chambers, 1 female in small chambers. We found that oviposition experiments using volatile compounds (i.e., fruit purées, acetic acid) required the large setups, which have increased air circulation compared to the medium and small setups. All other experiments (sugar-alone) were performed with medium setups, except for Fig 2A where all 3 setups were used, S2B Fig where small setups were used, and the dose–response experiments of Fig 2C and 2D, which were

performed in $12 \times 6 \times 4$ cm chambers with 5 females per chamber and a single 4-cm diameter Petri dish containing the oviposition substrate.

**Oviposition assays.** Flies were collected between 0 and 3 d after eclosion, aged in fresh food vials for 7 to 9 d, sorted on $CO_2$ just prior to experiments (except for dose–response experiments where they were sorted approximately 20 h prior), placed in behavioral chambers, and allowed to lay eggs in the chambers for 20 to 24 h in the dark. Eggs were counted manually and a preference index (PI) was calculated for each replicate (for two-choice experiments) using the following formula: PI = (# eggs on substrate 1 − # eggs on substrate 2) / total # eggs on both substrates. Chambers with less than 5 eggs were excluded from the PI calculation. Oviposition substrates were prepared by mixing the indicated compounds with a solution of separately boiled agar to achieve a final concentration of 0.5% (w/v) agar. Fruit purées were diluted to a final concentration of 30% (w/v). The sugar (glucose + fructose) concentration of undiluted ripe strawberry purée was measured to be approximately 5% (approximately 2.5% in S1A Fig, which shows a ½ dilution of ripe purée). The final sugar concentration of the ripe oviposition substrate (30% strawberry purée) is therefore approximately 1.6%. For acetic acid oviposition preference assays, a protein source (1% casein hydrolysate) was added to both substrates to stimulate egg laying, as acetic acid alone did not stimulate oviposition.

**Data analysis and statistics.** Analyses, statistics, and graphs were performed using custom scripts in Python (available in the supporting information file "S1 Python Scripts"). For two-choice experiments, the preference of each genotype/condition for one of the 2 substrates was assessed using a Mann–Whitney paired test comparing the # of eggs on substrate 1 versus # of eggs on substrate 2. $p$-Values <0.05 indicate a significant preference for one of the 2 substrates (filled circles in the graphs); $p$-values >0.05 indicate no significant preference for either substrate (open circles). To determine whether different genotypes/conditions showed different preferences in a given choice assay, their distributions of preference indices were compared using a Mann–Whitney U test ($p$-values indicated on the graphs). The distributions of total number of eggs laid were also compared using the Mann–Whitney U test. Dose–response experiments were analyzed using a Kruskal–Wallis H-test to determine whether each genotype/condition was significantly induced to lay eggs by the indicated sugar. The EC50 was determined as that where the egg-laying rate = (maximum rate − basal rate) / 2.

## Microscopy

*Dsuz-Gr64af-Gal4* expression was examined in live female legs and proboscises using tomato expression from our *UAS-GCaMP7s-T2A-Tomato* reporter line. Legs and proboscises were mounted in 1xPBS.

*Dsuz-* and *Dmel-Gr64af-Gal4* projection patterns in the brain (SEZ) were examined by immunostaining against tomato (using *UAS-Cd4tdTomato* reporters). Immunostaining was performed as described in [19] with the following antibodies: rabbit anti-RFP (Rockland, used at 1:1,000), mouse nc82 (Hybridoma bank, used at 1:20), secondary antibodies were anti-rabbit Alexa 488 and anti-mouse Alexa 647 (Rockland, used at 1:200). Brains were mounted in Slow-Fade medium (Thermo Fisher). Images were acquired using a Zeiss LSM780 confocal.

## Calcium imaging

In vivo calcium imaging was performed on 5- to 7-d-old mated females reared under the same conditions as for the behavioral experiments but starved for 24 h prior to the experiments. Flies were anesthetized on ice for 1 h, suspended by the neck on a Plexiglas block ($2 \times 2 \times 2.5$ cm) with the proboscis facing the center of the block, and immobilized using an insect pin (0.1 mm diameter) placed on the neck and fixed to the block with beeswax (Deiberit 502, Siladent,

209212). To prevent movement, the head was then glued to the block with a drop of resin (Gum rosin, Sigma-Aldrich −60895-, dissolved in 70% ethanol) so that the front of the head faced the microscope objective. The flies were then placed in a humidified box for 1 h to allow the resin to harden. A plastic coverslip with a small hole (diameter ~ distance between the 2 eyes) was placed on top of the head, fixed to the block with beeswax, and then sealed to the cuticle with two-component silicone (Kwik-Sil, World Precision Instruments), leaving the proboscis exposed to air below the coverslip. A drop of Ringer's saline (130 mM NaCl, 5 mM KCl, 2 mM $MgCl_2$, 2 mM $CaCl_2$, 36 mM sucrose, 5 mM HEPES (pH 7.3)) was applied on the head, and the cuticle covering the antennae area was then removed. To allow visual access to the anterior-ventral part of the SEZ, the trachea and fat body were removed and the gut was cut without damaging the brain or taste nerves. The exposed brain was then rinsed twice with Ringer's saline.

Imaging was performed using a Leica DM600B microscope with a 25x water objective. GCaMP7s fluorescence was excited using a Lumencor diode light source at 482 nm ± 25 and collected through a 505- to 530-nm bandpass filter. Images were acquired every 500 ms using a Hamamatsu/HPF-ORCA Flash 4.0 camera and processed using Leica MM AF 2.2.9.

Stimulation was performed by applying 140 μL of glucose, fructose, or sucrose (Sigma) diluted in water on the proboscis. Each experiment consisted of a recording of 100 prestimulation and 100 poststimulation images. The graphs show 30 prestimulation and 30 poststimulation images. For depolarization with 1 M KCl, the solution was applied directly to the brain.

Data processing was performed as described in [44]. For each image, ROIs were manually drawn on the left and right side of the SEZ in FIJI (https://fiji.sc/) and the signal intensity was quantified. These data were then manually inspected to exclude images with clear signs of drift and to select the side with the least variation in response. The initial intensity $F_0$ was calculated over 10 frames, 30 frames before the stimulus, and $\Delta F/F_0$ was expressed as a percentage. Peak $\Delta F/F_0$ was calculated as the average $\Delta F/F_0$ over 4 frames around the peak minus average $\Delta F/F_0$ over 4 frames immediately before the peak. Statistical differences in peak $\Delta F/F_0$ between species were assessed using the Mann–Whitney U test.

## Supporting information

**S1 Fig. Control experiments with the ripe and fermented substrates.** Controlled fermentation of a ripe strawberry purée with *Saccharomyces cerevisiae* and *Acetobacter pomorum* provides a model substrate for fermented fruit. **(A)** Glucose, fructose, acetic acid, and ethanol levels measured in g/L (mean + standard deviation) in the ripe and fermented purées before dilution in the oviposition substrates. Fermentation effectively depletes sugars and produces acetic acid and ethanol. **(B)** Mean final concentrations after dilution in the oviposition substrates. **(C)** Diet does not significantly alter the species' oviposition preferences for ripe vs. fermented substrates. The 2 species were allowed to develop on Nutrifly medium during the larval stages and either maintained on this medium during adulthood (black) or switched to standard cornmeal medium at eclosion (gray). *D. melanogaster* was also grown directly on standard medium during the larval stages (light blue), but we were unable to grow *D. suzukii* under these conditions. **(Left)** The oviposition preferences are independent of diet; **(right)** the total oviposition rate is significantly reduced for *D. suzukii* when aged on standard medium (*n* = 50, 30, 30, 50, 48). We therefore performed all experiments with flies reared on Nutrifly medium, which, unlike the standard medium, elicits an equivalent oviposition rate in both species. The data underlying this figure can be found in S1 Data.
(EPS)

**S2 Fig. Raw data from Fig 2A and 2E: Egg-laying rates on individual substrates in two-choice assays opposing varying sugar concentrations to plain agar. (A)** Raw data from Fig 2A: egg-laying rate on individual substrates from two-choice oviposition assays opposing 1.6% glucose + fructose to plain agar in 3 experimental setups of different sizes. For each replicate, the egg-laying rate is shown for the agar side (indicated by empty boxes at the bottom) and the sugar side (pink boxes). While *D. melanogaster* lays almost no eggs on the plain agar side, *D. suzukii* lays a significant number of eggs, resulting in a reduced strength of preference for sugar compared to *D. melanogaster*. **(B)** Oviposition preference in two-choice assays under conditions previously shown to induce sugar rejection in *D. melanogaster*. *D. melanogaster* avoids laying on sucrose and chooses plain agarose when acetic acid and ethanol are present on both sides in experimental setups corresponding to our small chambers [29,30]. We repeated these experiments with *D. melanogaster* and *D. suzukii*, opposing 200 mM sucrose to plain agarose in the presence of 0.8% acetic acid and 1.6% ethanol (orange boxes) on both sides, both substrates with a hardness of 1%, in small chambers. Consistent with published results, *D. melanogaster* rejects sucrose in the presence of acetic acid and ethanol, as does *D. suzukii*. However, both species oviposit on the sucrose side in the absence of acetic acid and ethanol, showing that sugar rejection under these conditions is due to an interaction with fermentation products. **(C)** Raw data from Fig 2E: egg-laying rate on individual substrates from two-choice assays opposing increasing concentrations of glucose (0.01% to 1%) to plain agar. The data underlying this figure can be found in S2 Data.
(EPS)

**S3 Fig. *Gr64af-Gal4*-positive neurons project in similar patterns in the CNS of *D. suzukii* and *D. melanogaster* and are not required for oviposition responses to acetic acid. (A)** Axonal projection patterns of the neurons labeled by the *Gr64af-Gal4* lines from *D. melanogaster* (top row) and *D. suzukii* (bottom row) in the CNS. A *UAS-Cd4tdTomato* reporter was used in both species, neuropil stained with nc82 antibody (magenta). *Gr64af-Gal4*-positive neurons project exclusively to the gustatory center—the sub-eosophageal zone (SEZ)—in both species. Merged images (far left) show z-projections over all planes. Images on the right and in the green channel-only show z-projections at different depths from anterior to posterior, revealing similar categories of arborization patterns in both species. **(B)** Two-choice oviposition assays opposing acetic acid at the indicated concentrations to plain agar, in the presence of 1% casein hydrolysate on both sides to stimulate egg-laying. Sugar-GRN inhibition does not abolish the response of either species to acetic acid ($n$ = 28, 29, 30, 30, 30, 30, 30, 30, 30, 25, 22, 18, 25, 22, 19, 27, 23, 18). The data underlying this figure can be found in S3 Data.
(EPS)

**S4 Fig. Additional information for calcium imaging experiments and controls for electrical hypersensitization via NaChBac. (A)** Individual traces for GCaMP experiments in the SEZ shown in **Fig 4A-4C. (B)** Peak $\Delta F/F_0$ data for sugar responses normalized to the average $\Delta F/F_0$ measured during stimulation with 1 M KCl. Normalized responses to glucose and sucrose are overall not significantly different between species. Normalized responses to 2.5% fructose are significantly lower in *D. suzukii* compared to *D. melanogaster* (Mann–Whitney U test, *p*-values indicated on the graph). *p*-Values in color below the graph indicate statistical comparisons with water controls. **(C, D)** Hypersensitization of sugar-GRNs in *D. melanogaster* via *UAS-NaChBac* expression increases responses to sugar. **(C)** Oviposition stimulation assays (i.e., no-choice) to increasing concentrations of glucose. Hypersensitizing *D. melanogaster* sugar-GRNs reproduces characteristic features of *D. suzukii*'s oviposition responses to sugar; it increases the egg-laying rate on plain agar and all sugar concentrations (*p*-values on the graph indicate comparison with both parental controls) but does not abolish the ability of the flies to adjust

their oviposition response to sugar concentration (Kruskal–Wallis test, shown on the right of the graph) ($n$ = 30 for each condition). **(D)** Expression of NaChBac in sugar-GRNs significantly increases calcium responses to 2.5% glucose, up to levels approaching 5% glucose stimulation in controls. Individual traces are shown in heat map (top) and peak $\Delta F/F_0$ distribution (bottom) ($n$ = 8, 9, 9). Data for 2.5% and 5% glucose controls are the same as in Fig 4B and 4C. The data underlying this figure can be found in S4 Data.
(EPS)

**S1 Dataset. Raw data for oviposition experiments displayed in Fig 1.** Egg-laying rate of each replicate, on each egg-laying substrate for two-choice assays (Fig 1A and 1C–1F) or on the single substrate for oviposition stimulation experiments (Fig 1B).
(XLSX)

**S2 Dataset. Raw data for oviposition experiments displayed in Fig 2.** Egg-laying rate of each replicate, on each egg-laying substrate for two-choice assays (Fig 2A, 2B, 2E and 2F) or on the single substrate for oviposition stimulation experiments (Fig 2C and 2D).
(XLSX)

**S3 Dataset. Raw data for oviposition experiments displayed in Fig 3D and 3E.** Egg-laying rate of each replicate, on each egg-laying substrate.
(XLSX)

**S4 Dataset. Raw data for GCaMP experiments displayed in Fig 4C and oviposition experiments displayed in Fig 4D and 4E.** Fig 4C: Peak GCaMP $\Delta F/F_0$ data for each brain for each condition and genotype. Fig 4D and 4E: Egg-laying rate of each replicate, on each egg-laying substrate of two-choice assays.
(XLSX)

**S1 Data. Raw data for S1 Fig.** S1A: glucose, fructose, acetic acid, and ethanol levels (g/L) measured in ripe (after ½ dilution in water) and fermented purées. S1B: final glucose, fructose, acetic acid, and ethanol levels (expressed as % w/v) in the oviposition substrates used in behavioral assays. S1C: Egg-laying rate of each replicate, on each egg-laying substrate of two-choice assays.
(XLSX)

**S2 Data. Raw data for oviposition experiments displayed in S2 Fig.** Egg-laying rate of each replicate, on each egg-laying substrate.
(XLSX)

**S3 Data. Raw data for oviposition experiments displayed in S3B Fig.** Egg-laying rate of each replicate, on each egg-laying substrate, for each genotype.
(XLSX)

**S4 Data. S4B: Raw data for GCaMP experiments displayed in S4B Fig: normalized peak $\Delta F/F_0$ data for each brain for each condition and genotype.** S4C: Egg-laying rate of each replicate for oviposition stimulation assay displayed in S4C Fig. S4D: GCaMP peak $\Delta F/F_0$ data for each brain of each genotype.
(XLSX)

**S1 PythonScripts. Custom Python scripts used to analyze oviposition assay data, run statistics, and make graphs.**
(PY)

## Acknowledgments

We are grateful to the Bloomington Drosophila Stock Center, Leopold Kurz, V. Courtier-Orgogozo, R. Benton, T. Auer, J. Blau and P. Andolfatto for fly stocks; D. Stern for the UAS-GCaMP7s-T2A-Tomato plasmid; Flybase for information support; J. Royet and A. Michelot for strains of *A. pomorum* and *S. cerevisae*, respectively.

## Author Contributions

**Conceptualization:** Matthieu Cavey, Yaël Grosjean, Benjamin Prud'homme.

**Data curation:** Martine Berthelot-Grosjean.

**Formal analysis:** Matthieu Cavey, Bernard Charroux, Solène Travaillard, Gérard Manière, Martine Berthelot-Grosjean, Sabine Quitard.

**Funding acquisition:** Matthieu Cavey, Yaël Grosjean, Benjamin Prud'homme.

**Investigation:** Matthieu Cavey, Bernard Charroux, Solène Travaillard, Gérard Manière, Martine Berthelot-Grosjean, Sabine Quitard, Caroline Minervino.

**Methodology:** Matthieu Cavey.

**Project administration:** Matthieu Cavey.

**Resources:** Matthieu Cavey, Brice Detailleur, Benjamin Prud'homme.

**Software:** Matthieu Cavey.

**Supervision:** Matthieu Cavey, Benjamin Prud'homme.

**Validation:** Matthieu Cavey.

**Visualization:** Matthieu Cavey.

**Writing – original draft:** Matthieu Cavey.

**Writing – review & editing:** Matthieu Cavey, Bernard Charroux, Gérard Manière, Martine Berthelot-Grosjean, Yaël Grosjean, Benjamin Prud'homme.

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
