## [Editor Report · Decision Letter 0]

6 Apr 2023

Dear Dr Cavey, 

Thank you for submitting your manuscript entitled "Increased sugar valuation in oviposition behavior of the fruit pest Drosophila suzukii" for consideration as a Research Article by PLOS Biology.

Your manuscript has now been evaluated by the PLOS Biology editorial staff, as well as by an academic editor with relevant expertise, and I'm writing to let you know that we would like to send your submission out for external peer review.

IMPORTANT: For the record, our "scooping policy" clock starts from the publication of Wang et al.'s eLife paper (18th November, 2022), not their preprint, so your submission on 31st March 2023 is well within the six-month protection period.

IMPORTANT: We think that your paper would be better considered as a Short Report. No re-formatting is needed, but please select "Short Reports" as the article type when you upload your additional metadata (see next paragraph).

Once your full submission is complete, your paper will undergo a series of checks in preparation for peer review. After your manuscript has passed the checks it will be sent out for review. To provide the metadata for your submission, please Login to Editorial Manager (https://www.editorialmanager.com/pbiology) within two working days, i.e. by Apr 11 2023 11:59PM.

Kind regards,

Roli Roberts

Roland Roberts, PhD

Senior Editor

PLOS Biology

rroberts@plos.org

---

## [Decision Letter · Decision Letter 1]

12 May 2023

Dear Dr Cavey,

Thank you for your patience while your manuscript "Increased sugar valuation in oviposition behavior of the fruit pest Drosophila suzukii" was peer-reviewed at PLOS Biology. It has now been evaluated by the PLOS Biology editors, an Academic Editor with relevant expertise, and by four independent reviewers. 

In light of the reviews, which you will find at the end of this email, we would like to invite you to revise the work to thoroughly address the reviewers' reports.

IMPORTANT: You'll see that all four reviewers are very positive about your study. However, they do raise a number of concerns that should be addressed. Specifically, there are some recurrent experimental requests that would improve your study; three of the reviewers ask that you explore the role of acetic acid, and two suggest using Ca2+ imaging, which we believe you already have set up. I discussed these requests with the Academic Editor, who thought that these were reasonable and would be very helpful, but added "Looking for temporal dynamics differences is asking for too much though. The point that a couple of them raised about being wary about assessing Ca2+ differences w/o taking into consideration basal levels, GCaMP fluorescence levels etc is a pretty important one that the authors should really carefully validate." The Academic Editor also wanted you to do a better job (in your Introduction) of describing the Wang/Carlson findings, saying "There is almost nothing mentioned in the paper about decreased expression of Grs that the Carlson lab found and increased expression of the mechanosensory channel NompC (unless I missed it). Those should be mentioned and addressed."

Given the extent of revision needed, we cannot make a decision about publication until we have seen the revised manuscript and your response to the reviewers' comments. Your revised manuscript is likely to be sent for further evaluation by all or a subset of the reviewers.

**IMPORTANT - SUBMITTING YOUR REVISION**

*Re-submission Checklist*

*Published Peer Review*

*PLOS Data Policy*

*Blot and Gel Data Policy*

Sincerely,

Roli Roberts

Roland Roberts, PhD

Senior Editor

PLOS Biology

rroberts@plos.org

REVIEWERS' COMMENTS:

Reviewer #1:

In the manuscript entitled "Increased sugar valuation in oviposition behavior of the fruit pest Drosophila suzukii", Cavey et al. investigate the behavioural and physiological basis of niche adaptation (via oviposition) to ripe fruit in D. suzukii. 

Complementing previous studies, they show that sugar is a major driver for Dsuz choice to oviposit on ripe vs. fermented fruit which is preferred by D. melanogaster. By developing novel genetic tools in Dsuz, they can directly test the behavioural importance of sugar sensing neurons for oviposition via silencing experiments, and they are able to perform Calcium imaging experiments in sugar sensing gustatory receptor neurons. 

Technical merit and experimental design: The manuscript is well written, nicely illustrated and the presented data supports the conclusion of the authors. Experiments are well designed, controlled and technically well done. The novel genetic tools introduced for Dsuz represent a major effort and will open exciting new research opportunities in the future. 

Novelty and significance: One could argue that many of the described experiments overlap with a recent publication using the same experimental system [23]. However, I think the complementary genetic approach taken by the authors, the careful dissection of sugar induced oviposition and the emerging hypothesis of central brain changes merit publication in Plos Biology. Especially the potential role of central brain circuits in behavioural adaptation is very exciting and will be a promising avenue for future studies the authors are well positioned to perform. This observation is of interest not only to Drosophila researchers but other (neuro)scientist interested in brain function and evolution. 

Prior to publication however, I would like the authors to consider a couple of points: 

A) Several recent studies using Calcium imaging in Drosophila peripheral sensory neurons have described ON and OFF responses upon stimulus presentation and removal (e.g. PMID 34197729 in sweet neurons). Given the tools the authors have available, I would love to see if there are some species-specific temporal dynamics in the shown sugar responses that could impact central interpretation of peripheral signals. 

B) The authors nicely show the species separation between Dmel and Dsuz when offering ripe or fermented fruit (Figure 2D) and that Dsuz choice is mainly driven by sugar content. Given the chemical analysis in Sup. Fig. 1, I wonder if the authors could add acetic acid to the Ripe substrate and make Dmel prefer this combination over fermented. This experiment would allow to disentangle the relative role of acetic acid and sugars for both species. In Dsuz, even though they prefer oviposition of acid containing substrate (Fig. 3E), the sugar input seem to be more important (contrary to Dmel). 

Minor points: 

Figure 1A: Could the authors plot the number of eggs per species laid? They mention several times in the manuscript that Dsuz lays eggs on agar only (and it is depicted in Fig. 1C) but it could help to show this once at the beginning of the manuscript

Figure 1C: Please add strain labels

Line 82: delete one "increase(d)"

Line 123 : "D. suzukii does not discriminate concentrations ≥ 1% glucose, whereas D. melanogaster always shows preference for the higher concentration": This phrasing is a a bit misleading as there is a preference in Dsuz at higher concentrations and it prefers 2.5% sugar vs. plain agar. Could the authors please rephrase this sentence? 

Line 181: "We therefore developed a controlled- fermentation protocol to produce a chemically-defined, sugar-free fermented substrate using industrial strawberry purée as a starting ripe substrate (S1A Fig) ": I recommend adding here that the authors analysed the chemical composition and this is actually shown in Fig. S1A

Reviewer #2:

The submitted manuscript delves into the effects of sugar in the oviposition substrate preference of two fruit fly species: Drosophila suzukii (Dsuz) and D. melanogaster (Dmel). While Dsuz lays its eggs on ripe fruit, Dmel prefers fermenting ones. Sugar appears to be a complicated cue in Dmel, as it can have a positive or negative effect depending on various factors. The authors have taken care to control as many variables as possible, resulting in an extensive set of experiments that can be somewhat challenging to summarize in a concise manner. In essence, the authors found that Dsuz and Dmel differ in their assessment of sugar when it comes to egg-laying. Dsuz is more sensitive to sugar than Dmel and prefers substrates with up to 1.6% sugar. However, Dsuz responds to lower sugar concentrations than Dmel, indicating that it may be more sensitive to sugar. Surprisingly, the authors found that Dsuz has fewer sugar-sensitive GRNs that are less sensitive to sugar. 

General comments: 

The paper is well-written, and the experiments presented are done to a high standard. The results are, however, quite confusing, and I'm not sure how much wiser I'm about this issue after having read this manuscript. There is an obvious need for more work here, and not only in Dsuz, but also in Dmel, to figure out how sugar affects egg-laying behavior. Despite the complexity, and sometimes even counterintuitive nature of their findings, the authors present their data clearly and thoughtfully, and also provide some reasonable arguments that explain some of the odder findings, which make this research noteworthy. 

Specific comments: 

Line 59-62: Most Drosophila do not feed on rotting fruit. Drosophila (i.e. Sophophora and Drosophila) occupy a wide range of niches, which of these is the most common we do not know, but fruit is unlikely to be the most prevalent breeding substrate. As for utilizing ripe fruit, and its novelty within Drosoohilidae, again, this is not known. 

Line 169-171: Unclear. I don't understand what the authors mean. In fact this whole section is a bit hard to follow. 

Figure 2E: no stats reported. 

Line 211ff: Wouldn't the Orco mutants previously generated by these authors been a good tool to use here? It would quite clearly demonstrate whether sugar is indeed the main driver of this behavior. 

Figure 3C: The PI of the Dsuz Gr64af-Gal4>UAS-Kir2.1 line is not all that different from parental Gr64-gal4 control. Is that really significant, if so, must be tough and go. Moreover, the PI (ca 0.25) is roughly about the same as the PIs seen in Figure 1A. 

Line 269: Does Gr64af-Gal4 really label all sugar neurons? Please rephrase. 

Figure 3D: The PI of Dmel CS looks to be not significantly different from 0. And also different from the PI in Figure 2C. 

Figure 3E: But there is an effect at 0.1%. Any explanation?

Line 280ff: Again why not use the Orco-Gal4? 

Line 336ff. This section is rather confusing. 

Figure 4F: Curious that the addition of sugar to ripe fruit overrides the innate attraction to fermentation volatiles. I think this observation warrants some sentences. 

Line 411ff: But…Dmel likes sugar, the more the merrier it seems. How does that fit with Dmel having inhibitory sugar GRNs, whereas Dsuz, that do not like high sugar concentrations, lack these?

Reviewer #3:

This paper by Cavey et al. addresses how different Drosophila species have evolved to occupy different ecological niches, a useful model for studying evolutionary adaptations more broadly. D. suzukii prefers to lay eggs on ripe fruit, which is high in sugar, whereas D. melanogaster prefers fermented fruit, which contains less sugar. Does sugar perception contribute to this behavioral difference? The authors make three major findings. First, sugar stimulates egg-laying more strongly and at lower concentrations in D. suzukii than in D. melanogaster. In other words, D. suzukii is more responsive to sugar, despite the fact that it shows lower sugar preference in a two-choice assay (due to high basal levels of egg-laying). Second, sugar perception is required for D. suzukii to show preference for ripe fruit puree. Third, despite D. suzukii's increased behavioral responsiveness to sugar, it has fewer sugar neurons than D. melanogaster and the responses of those neurons are weaker. Thus, the increased behavioral response likely arises from increased sensitivity in the downstream circuit. 

The experiments are thorough and rigorous, and the results are nicely synthesized. I particularly appreciated how the authors developed a controlled paradigm for fruit fermentation and how they dissociated sugar stimulation of egg-laying from two-choice sugar preference. The development and use of genetic tools for neuronal imaging and manipulation in a non-model species is also commendable. I would support publication of this study as a Short Report, provided that my comments below can be addressed.

Major comments:

1) The title and several portions of the text refer to "increased sugar valuation" in D. suzukii. I do not agree with this terminology. "Value" and "valuation" are technical terms in certain fields of neuroscience that have a specific meaning, which I do not think is appropriate for this study where it seems like "value" simply refers to the ability of sugar to drive egg-laying behavior and choice. 

2) Based on Figure 2D, the sugar concentration is an important factor driving D. suzukii's preference for ripe fruit, but is NOT an important factor driving D. melanogaster's preference for fermenting fruit. The importance of sugar perception for D. suzukii was verified by Kir silencing of sugar GRNs. Was a similar experiment performed for D. melanogaster (silencing sugar GRNs while testing preference for ripe vs. fermented puree)? The expectation would be that there is not a significant effect. I think the fact that two species use different cues to choose between the same two substrates (and make opposing choices) is very interesting and could be emphasized more, along with a description of the other cues that could be involved (e.g. acetic acid, ethanol). 

3) In Figure 4, was calcium imaging performed for other sugars besides glucose? Seeing similar results with other sugars such as sucrose and fructose would greatly strengthen the claim that neurons in D. suzukii show weaker responses, different temporal dynamics, and a flatter dose-response curve. 

4) For the imaging data, I would be careful when comparing the strength of neuronal responses across species. Even within a species, neuronal responses can vary between driver lines. Ideally using deltaF/F should normalize for differences in transgene expression, but I don't think this always is completely true, e.g. if baseline values are close to the detection threshold or maximal response values are at a ceiling. I do appreciate that the authors didn't overstate this claim and focused on comparing the shape of the dose-response curves and the temporal dynamics. 

Minor comments:

1) For readers who are familiar with previous studies investigating egg-laying aversion to sugar in D. melanogaster (e.g. Yang et al., 2008; Vijayan et al., 2022) it might be helpful to add a sentence mentioning how sugar can be either attractive or aversive for oviposition (and potentially why?). 

2) Reference 23 was published in eLife last year; please update the citation.

3) In Figure 1A, it would be helpful to add text labels to the schematics so it's clear where the oviposition substrates are and what the circular thing in the center is. 

4) The text mentions that the lab-controlled fermented substrate, starting from strawberry puree, is "sugar-free". I was initially confused about whether this means no additional sugar has been added or whether the natural sugars were somehow removed. Based on later experiments I assume the former, but this should be clarified. 

5) In Figure 2C, it would be nice to show statistics to know which species/strains have a preference that's significantly different from zero (e.g. a one-sample t-test).

6) Can the authors speculate on why D. suzukii has a lower number of sugar GRNs, since this is counterintuitive based on their increased oviposition response to sugar? Could it be related to any known differences in feeding behavior? 

7) On line 385, I don't think it's fair to characterize Wang et al.'s conclusion as saying "sugar was not an important cue for D. suzukii's oviposition decisions". They suggested that taste cues, including sugar, are less important than in D. melanogaster, but that is not the same as "not important". 

8) On line 390 of the discussion, the authors state, "D. suzukii's overall weaker preference for sugar in two-choice assays is most likely due to a high basal egg-laying rate on plain agar". This is a very important point that could have been made more clearly in the results section.

Reviewer #4:

Certain Drosophilids can contribute to substantial agricultural loss, most prominently the Asian species Drosophila suzukii, which has rapidly expanded in Europe and Americas. 

A striking difference in the ecology of D. suzukii is that these animals lay their eggs in ripe fruits as compared to rottening fruits in other Drosophila species. The neuronal and evolutionary context of how this preference emerged and is controlled remains largely unknown. One difference between the two food types - ripe versus fermented- is the content of sugars and the processing of these substances. In the current manuscript Cavey and colleagues investigate the relevance of different sugars, fresh fruits and fermented fruits for oviposition choice in D. suzukii. 

They first show that compared to melanogaster, which prefers sugar D. suzukii showed a lower preference to sugars as compared to D. melanogaster (in a "choice assay"; however both species show preference for sugars (in a "non-choice assay"); The authors also show that D. suzukii is "unique" in ripe fruit preference among Drosophilds.

Since in D. melanogaster Gr64 genes are involved in sugar preference, the authors tested next if the Gr64 expressing neurons are also responsive to ripe fruit. They used behavioral and Calcium imaging experiments to corroborate this notion.

While genetic tools in D. melanogaster are highly advanced and abundant, the same is not yet true for other Drosophilids. The generation of GCaMP and other tools (UAS-KIR, Gr64-Gal4) used here are elegant and powerful. It is evident that this is a major achievement. 

Overall, this is an elegant manuscript addressing a critical question in the largely unexplored biology of D. suzukii. I feel particularly the technical advances with this model is impacting. 

There are a few points that I feel would be beneficial to be addressed.

- Above all, the preference of D. suzukii to ripe fruits as compared to decomposing fruits is among the best-known facts about this species; also making it an actual pest for certain types of ripe fruits. It seems intuitive that there must be some difference between the ripe and fermenting fruits, that are differentially "attractive" between D. suzukii and e.g. D. melanogaster. The authors here propose that perception of sugars to be such an aspect. The provided data indeed are in agreement to this notion, however much remains unclear how this sweet-perception difference might contribute to the difference in behavior/ecology. The only point where this is taken up to some degree is fig suppl 1.; in which some glucose, fructose, acetic acid and ethanol differences between ripe and fermenting fruits are measured. I feel at the very least this should be more discussed as it provides a "real-world" measurement and provides a possible basis of why animals chose differently. 

Given the context its hard as a reader to feel that the context is to some degree not satisfying. Integrating behavioral experiments with acetic acid or ethanol and the combination with sugars might give at least partly explanations. I feel such experiments would provide some more context. 

- The reduced GCaMP response in D. suzukii could simply be due to GCaMP expression levels or activity of Gal4/UAS. Providing a positive response (e.g. high KCl or similar) would provide evidence that this is not an artifact.

- In ripe fruit preference also D. simulans and willistoni show preference for ripe fruit. I wonder if the authors have a view or explanation on this observation.

---

## [Decision Letter · Decision Letter 2]

25 Oct 2023

Dear Dr Cavey,

Thank you for your patience while we considered your revised manuscript "Increased sugar valuation in oviposition behavior of the fruit pest Drosophila suzukii" for publication as a Short Report at PLOS Biology. This revised version of your manuscript has been evaluated by the PLOS Biology editors, the Academic Editor and the original reviewers.

Based on the reviews, we are likely to accept this manuscript for publication, provided you satisfactorily address the following data and other policy-related requests.

IMPORTANT - please attend to the following:

a) Please could you change your title to something more accessible to our broader readership, and with an active verb? We suggest "Increased sugar valuation drives egg-laying behavior in the fruit pest Drosophila suzukii" (unless you think that the verb "drives" is too strong, in which case you can choose something softer, like "contributes to").

b) Please address my Data Policy requests below; specifically, we need you to supply the numerical values underlying Figs 1ABCDEF, 2ABCDEF, 3DE, 4CDE, S1ABC, S2ABC, S3B, S4BCD, either as a supplementary data file or as a permanent DOI’d deposition.

c) Please cite the location of the data clearly in all relevant main and supplementary Figure legends, e.g. “The data underlying this Figure can be found in S1 Data” or “The data underlying this Figure can be found in https://doi.org/10.5281/zenodo.XXXXX”

d) Please make any custom code available, either as a supplementary file or as part of your DOI'd deposition.

We expect to receive your revised manuscript within two weeks. 

*Published Peer Review History*

*Press*

Sincerely,

Roli Roberts

Roland Roberts, PhD

Senior Editor,

rroberts@plos.org,

PLOS Biology

DATA POLICY:

Regardless of the method selected, please ensure that you provide the individual numerical values that underlie the summary data displayed in the following figure panels as they are essential for readers to assess your analysis and to reproduce it: Figs 1ABCDEF, 2ABCDEF, 3DE, 4CDE, S1ABC, S2ABC, S3B, S4BCD. NOTE: the numerical data provided should include all replicates AND the way in which the plotted mean and errors were derived (it should not present only the mean/average values).

CODE POLICY

Per journal policy, as the code that you have generated is important to support the conclusions of your manuscript, we require that you make it available without restrictions upon publication. Please ensure that the code is sufficiently well documented and reusable, and that your Data Statement in the Editorial Manager submission system accurately describes where your code can be found.

DATA NOT SHOWN?

REVIEWERS' COMMENTS:

Reviewer #1:

The authors addressed all points to my full satisfaction. I support publication in its current form.

Reviewer #2:

The authors have addressed my concerns, and I see no need for further rounds of revisions. 

Reviewer #3:

I have reviewed the revised manuscript. I believe the paper has been significantly improved and that my comments and those of the other reviewers have been sufficiently addressed. I have no further concerns.

Reviewer #4:

I feel all points raised have been addressed. It is an intriguing report containing several relevant pieces of information.

---

## [Editor Report · Decision Letter 3]

15 Nov 2023

Dear Dr Cavey,

Thank you for the submission of your revised Short Report "Increased sugar valuation contributes to the evolutionary shift in egg-laying behavior of the fruit pest Drosophila suzukii" for publication in PLOS Biology. On behalf of my colleagues and the Academic Editor, Piali Sengupta, I'm pleased to say that we can in principle accept your manuscript for publication, provided you address any remaining formatting and reporting issues. These will be detailed in an email you should receive within 2-3 business days from our colleagues in the journal operations team; no action is required from you until then. Please note that we will not be able to formally accept your manuscript and schedule it for publication until you have completed any requested changes.

Sincerely, 

Roli Roberts

Senior Editor

PLOS Biology

rroberts@plos.org